# Influence of Magnesium Ions on the Preparation and Storage of DNA Tetrahedrons in Micromolar Ranges

**DOI:** 10.3390/molecules24112091

**Published:** 2019-06-01

**Authors:** Yue Hu, Zhou Chen, Zheng Hou, Mingkai Li, Bo Ma, Xiaoxing Luo, Xiaoyan Xue

**Affiliations:** Department of pharmacology, Fourth Military Medical University, Xi’an 710000, China; yuehu918@163.com (Y.H.); chenzhou_cky@163.com (Z.C.); hzh_0001@163.com (Z.H.); mingkai@fmmu.edu.cn (M.L.); mbcarl@163.com (B.M.)

**Keywords:** DNA tetrahedron, self-assembly, biomaterials, nanoparticles, magnesium ions, lyophilization

## Abstract

The DNA tetrahedron (Td), as one of the novel DNA-based nanoscale biomaterials, has been extensively studied because of its excellent biocompatibility and increased possibilities for decorating precisely. Although the use of Td in laboratories is well established, knowledge surrounding the factors influencing its preparation and storage is lacking. In this research, we investigated the role of the magnesium ions, which greatly affect the structure and stability of DNA. We assembled 1, 2, 5, 10 and 20 μM Td in buffers containing different Mg^2+^ concentrations, demonstrating that 2 and 5 mM Mg^2+^ is optimal in these conditions, and that yields decrease dramatically once the DNA concentration reaches 20 μM or the Mg^2+^ concentration is lower than 0.5 mM. We also verified that the Td structure is retained better through freeze-thawing than lyophilization. Furthermore, a lower initial Mg^2+^ (≤2 mM) benefited the maintenance of Td structure in the process of lyophilization. Hence, our research sheds light on the influence of Mg^2+^ in the process of preparing and storing Td, and also provides some enlightenment on improving yields of other DNA nanostructures.

## 1. Introduction

Recently, the DNA nanostructure has attracted a lot of attention because of its special properties, such as flexible size and shape, excellent biocompatibility, and increased possibilities for decorating precisely. Stabilized by strong hydrogen bonds, the DNA nanostructure can easily self-assemble into designed shapes, leading to its wide application in diagnosis, imaging or drug delivery [1]. Among these nanoparticles, the DNA tetrahedron (Td), which has a high yield and rigid structure, is widely used [2]. It has been reported that a Td can be internalized by a cell even without transfection reagent [3], based on which scientists have designed various smart systems to deliver CpG motifs [4], siRNA [5] or small molecular drugs [6], and to detect mRNA in living cells [7]. However, there are still many problems that need to be explored for the further development of the Td, including factors influencing a Td’s assembly process and its stability during storage.

Previously published data showed that the types and concentrations of cations in the solution are the key factors affecting the stability of the DNA structure. Compared with monovalent cations, divalent cations, such as Mg^2+^, have stronger interactions with DNA and a greater influence on DNA structure [8]. There are two types of interactions between DNA and Mg^2+^: electrostatic force (phosphate groups) and chemical bonds (like hydrogen bonds and coordinate bonds) [9]. In a solvent, Mg^2+^ has a coordination shell with six water molecules that can bind to the O6 of the DNA to form hydrogen bonds. Also, in some cases, coordinate water molecules are replaced by the N7 and O6 sites of the DNA guanine, forming coordination bonds between the DNA and Mg^2+^. The effects of magnesium can be considered as ‘protective effects’ of DNA, not only decreasing the intramolecular repulsion in DNA but also making the DNA duplex more rigid [10]. Importantly, it has been confirmed that the quantity of magnesium ions affects DNA structure. DNA is stabilized in the B conformation when Mg^2+^ is at a modest concentration, but when the concentration is extremely high, the DNA structure exhibits a more compact form [11].

Therefore, the influence of Mg^2+^ on DNA nanostructure in applicable conditions has been studied. It was reported that a low concentration of Mg^2+^ (≤1 mM) is detrimental to the stability of DNA nanostructures [12]. However, Keller’s group proposed that high Mg^2+^ concentrations are not necessary for maintaining stability if an optimal buffer is chosen based on the specific form of the DNA nanostructure [13]. Furthermore, there are other ways to ensure DNA nanostructure integrity against the impacts of Mg^2+^, such as oligolysine-coated protection [14]. However, the role of Mg^2+^ in DNA nanostructure preparation and storage (such as freezing-thawing and lyophilization) has not yet been investigated; such studies are important for improving experimental reproducibility or reducing batch variations through the preparation of one batch of Td used in multiple rounds of experiments for a longtime.

In this study, we synthesized Td at different Mg^2+^ concentrations and clarified the influence of Mg^2+^ on Td preparation and storage, which provides some enlightenment toward improving yields of other DNA nanostructures.

## 2. Results

Most previous studies ([4], [15], [16], and [17]) reported that a Td could be constructed in a Tris-Mg buffer (TM buffer) with 5 or 50 mM Mg^2+^; therefore, we first assembled the Td at varied concentrations (1, 2, 5, 10, and 20 μM) in the common TM buffers containing 5 or 50 mM Mg^2+^. The sequences of the DNA strands are presented in Appendix A. The yields and morphologies of the tetrahedrons were identified by electrophoresis and atomic force microscope (AFM), respectively. The results show that in the buffer containing 5 mM Mg^2+^, Td was successfully assembled at all concentrations except for 20 μM. The band of the 20 μM DNA sample was stagnant in the well of gel, indicating the aggregation of DNA strands under this condition; these results correspond with the images observed by AFM (Figure 1). Nevertheless, under the condition of 50 mM Mg^2+^, the Td could only be formed in the first two DNA concentrations (1 and 2 μM), while the samples with the other concentrations (5, 10 and 20 μM) exhibited diameters that were much larger than normal (Figure 1). These results indicate that the concentrations of Mg^2+^ had an influence on the self-assembly process of the Td and that high Mg^2+^ concentrations may be detrimental to the formation of high-concentration Td.

Next, we further clarified the role of Mg^2+^ in the self-assembly process of Td by synthesizing Td in TM buffers containing diverse Mg^2+^ concentrations (0.05, 0.5, 2, 10 and 25 mM MgCl_2_). The electrophoresis results (Figure 2A) show that different concentrations of Td were assembled in 2 mM MgCl_2_. However, 1, 2 or 5 μM Td could be assembled in 25 mM Mg^2+^, and aggregation appeared when the concentration of DNA increased to 10 or 20 μM. Nevertheless, the samples of 10 or 20 μM prepared in lower Mg^2+^ density (0.5 and 0.05 mM) also showed poor yields, and bands with faster mobility were observed in the electrophoresis results, indicating the existence of free DNA strands. The dynamic light scattering (DLS) results further confirm that the successfully prepared Td had a hydrodynamic size of ~12.36 nm, and when aggregation occurred, the particle sizes increased with an increase of Mg^2+^ concentration (Figure 3). The polydispersity index (PDI) values of products are listed in Appendix A, indicating that greater particle heterogeneity as aggregation occurred. (Only particles larger than 3 nm can be measured in this apparatus; therefore, the DLS results of the Td prepared in 0.05 or 0.5 mM Mg^2+^ were not available.)

After estimating the yields of Td in various environments by gray scanning (Appendix A and Figure 2B), we found that adequate Mg^2+^ intensity is the first condition for constructing high-yielding Td; this observation was based on the fact that yields decreased dramatically to less than 20% in 0.05 mM Mg^2+^, no matter what concentration of DNA nanostructures was prepared. Notably, DNA in higher concentrations is more vulnerable to the change in Mg^2+^ density. As illustrated in Appendix A, when the concentration of Mg^2+^ changed from 2 mM to 25 mM, the yield of 20 μM Td slumped from 55.4% to 7.8%, while that of the 1 μM sample decreased from 77.53% to 52.22% (Appendix A). 

As for the Td storage, we examined the effect of Mg^2+^ during low-temperature freezing and vacuum drying. In this part, we firstly prepared Td in TM buffer containing 5 or 2 mM Mg^2+^, and then carried out either the process of freeze-thawing three times or the process of lyophilization treatment. As a result, after freeze-thawing treatment, irrespectively of time, the bands of Td prepared under both 5 and 2 mM Mg^2+^ conditions had an identical mobile speed, and the gels showed no difference to that of freshly prepared ones (Figure 4). These results indicate that iterative freeze-thawing processing had no influence on the structure of the Td. However, when dissolving the freeze-dried powder of the tetrahedrons in an equivalent volume of deionized water, the tetrahedrons prepared in 5 mM Mg^2+^ mostly stayed in wells and formed disordered aggregations, as observed on the AFM images. Differently, tetrahedrons at all concentrations prepared in 2 mM Mg^2+^ mostly maintained their previous structures (Figure 5). These results indicate that the stability of the Td structure was more sensitive to the process of vacuum drying than to that of low-temperature freezing. To further verify the influence of Mg^2+^ in the process of lyophilization, we decreased its concentration to 0.5 mM during preparation, repeated the experiment, and found that the Td after the freeze-drying in this condition produced little aggregation (Figure 5). All these data imply that lower Mg^2+^ in the initial prepared buffer benefits the structure stability of the Td in the process of lyophilization. 

Because PBS, Mueller-Hinton broth (MHB) and Dulbecco’s modified eagle’s medium (DMEM) are commonly-used biological buffers for cellular or bacterial culture, we investigated whether lyophilizated powder of Td could be directly dissolved in these buffers. In this study, freeze-dried powders of 1 or 5 μM Td, prepared in TM buffer containing 5, 2, or 0.5 mM Mg^2+^, were dissolved in different solutions. We found that aggregation was more pronounced in the samples dissolved in PBS, MHB and DMEM than the sample dissolved in water (Figure 6A). The results indicate that any solvent containing more complex components than water resulted in the aggregation of Td powder after re-dissolving. Furthermore, we investigated the feasibility of concentrating Td (prepared in 2 mM Mg^2+^) by dissolving its freeze-dried powder in different volumes of deionized water, which were 3/4, 1/2 or 1/4 of the initial volume of prepared buffer. As demonstrated by the electrophoresis results (Figure 6B), the Td powders were successfully re-dissolved and mostly retained their original structures at the above volumes, indicating that it is a practicable way to concentrate Td prepared in an appropriate Mg^2+^ intensity (2 mM Mg^2+^). 

## 3. Discussion

These results imply that Mg^2+^ concentration plays an important role when synthesizing Td, and that relatively low concentrations of Mg^2+^ are more helpful in producing Td with a high concentration; however, the formation will not succeed without enough Mg^2+^. In addition, the yield declined visibly when the Td concentration increased to 20 μM (Figure 7A). The exact quantitative balance of DNA and Mg^2+^ and the underling mechanism are still unclear. It would be meaningful to research the exact interaction between DNA and Mg^2+^ in the process of assembling tetrahedrons, which will guide further studies to meet the requirements of diverse DNA nanostructures. Furthermore, we found that vacuum drying exerts a more conspicuous effect on the Td structure than freeze-thawing. Notably, the Mg^2+^ concentration during Td preparation has a great influence on this process (Figure 7B). In addition, considering the complexity of the lyophilization process and for avoiding the interference of other components, water as a re-dissolving solution is the best choice. It has been reported that high Mg^2+^ can cause condensation between DNA double helixes by ion-bridging, and this process varies according to different kinds of DNA configuration [18], which may explain our observed results that more Mg^2+^ tends to result in unexpected aggregation during the preparation and lyophilization. When preparing the Td, the unexpected entanglement between DNA strands is more likely to occur due to high DNA concentration, which may lead to DNA aggregation. However, the prepared Td remains its structure during lyophilization only if the Mg^2+^ concentration in the prepared buffer is appropriate because hydrogen bonds in the Td are beneficial for maintaining the structure. As for the influence of the buffer when re-dissolving the freeze-dried powder, we speculate that the bond angle is distorted due to the special structure of Td, which is highly vulnerable to complicated elements in PBS, MHB or DMEM when dissolved. Importantly, our results suggest that freeze-dried dissolving is a practicable way to concentrate Td if it is prepared in initial solutions with less than 2 mM Mg^2+^. Nevertheless, the reaction mechanism during this process is not clear. It would be valuable to further study the relationship between Mg^2+^ and DNA and to seek the conformations of the crystal structure of the Td during the process.

## 4. Materials and Methods

### 4.1. Materials

All the oligonucleotides were obtained from Sangon Biotech (Shanghai, China). The 2000 bp DNA marker was bought from Takara (Tokyo, Japan). Magnesium chloride hexahydrate was purchased from Tian Li (Tianjin, China). Trizma base was obtained from Sigma-Aldrich (St. Louis, MO, USA). Agarose was bought from Lonza (Rockland, ME, USA). PBS, Mueller–Hinton broth (MHB) and DMEM were purchased from Boster (Wuhan, China), Land Bridge (Beijing, China) and Life technologies (Waltham, MA, USA), respectively. The water used in all experiments was prepared via a Millipore Milli-Q purification system with a resistivity higher than 18 MΩ cm-1.

### 4.2. Methods

#### 4.2.1. Preparation of DNA Tetrahedron

Four DNA strands in equal molar quantities were mixed in TM buffer (12.5 mM Tris, 0.05/0.5/2/5/10/25/50 mM MgCl_2_, pH = 7.8-8.0), heated to 95 °C for 5 min with MJ MiniTM 48-Well Personal Thermal Cycler, and then rapidly cooled on ice for at least 1 h.

#### 4.2.2. Agarose Gel Electrophoresis

One percent (w/v) agarose gel was prepared by adding EtBr. An amount of 5 μL prepared DNA tetrahedron mixed with a loading buffer was loaded and run at 95 V for 35 min in an ice bath.

#### 4.2.3. AFM Imaging

All samples were diluted to 2.5 nM in TM buffer. An amount of 10 μL DNA tetrahedron was dropped onto a freshly cleaved mica and incubated for 5 min to achieve strong absorption onto the surface. Then, the mica was rinsed with filtered deionized water and dried gently with compressed nitrogen. After that, the samples were scanned in the tapping mode on the Agilent 5500 SPM. 

#### 4.2.4. Freeze-thawing of DNA Tetrahedron

The DNA tetrahedron was frozen to −80 °C for 12 h and thawed at room temperature, repeatedly. The samples were collected immediately after every circle of freeze-thawing.

#### 4.2.5. Lyophilization of DNA Tetrahedron

The tubes containing DNA tetrahedron were sealed with holey sealing films, and frozen to −80 °C for 2 h. Then, the samples were placed into a FD5-3 Freezer Dryer, which had been cooled to −50 °C ahead of time and maintained a vacuum until the nanoparticles transformed into powder. 

#### 4.2.6. Estimation of Td Yield

Using the software ImageJ, the band intensity of Td was compared with the total intensity of the corresponding lane. This ratio was determined as the Td yield, which was obtained from three independent experiments.

## 5. Conclusions

In this study, we took Td as an example, investigating the role of Mg^2+^ concentration in Td preparation and storage. Firstly, we assembled the Td in buffers containing different Mg^2+^ concentrations and we found that 2 or 5 mM Mg^2+^ to be the optimal condition, and that at higher or lower levels the yields of Td will reduce. Furthermore, we also verified the viability of maintaining the structure of Td by the process of freeze-thawing rather than lyophilization, and confirmed that the concentration of Mg^2+^ was important for the structural stability of Td during lyophilization.

## Figures and Tables

**Figure 1 molecules-24-02091-f001:**
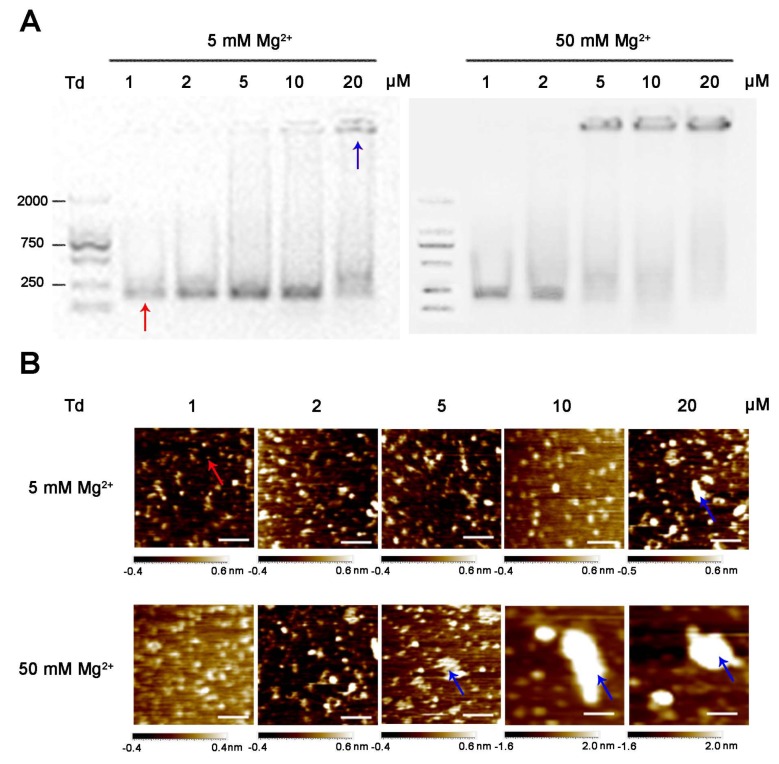
Characteristics of tetrahedrons (Tds) (1, 2, 5, 10 and 20 μM) prepared in 5 and 50 mM Mg^2+^. (**A**) Agarose gel of DNA products. (**B**) AFM images of DNA products. Scale bars: 100 nm. Red and blue arrows represent successfully assembled Td and aggregated DNA, respectively.

**Figure 2 molecules-24-02091-f002:**
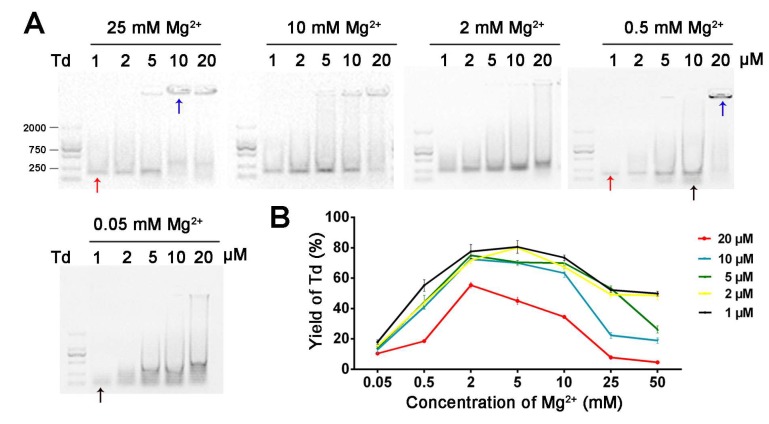
The influence of Mg^2+^ and Td concentrations on preparing Td in micromolar ranges. (**A**) Agarose gel of 1, 2, 5, 10 or 20 μM DNA products under various Mg^2+^ concentrations. (**B**) Yield curve of Td versus Mg^2+^ and Td concentrations. Red, blue and black arrows represent successfully assembled Td, aggregated DNA and free DNA strands, respectively.

**Figure 3 molecules-24-02091-f003:**
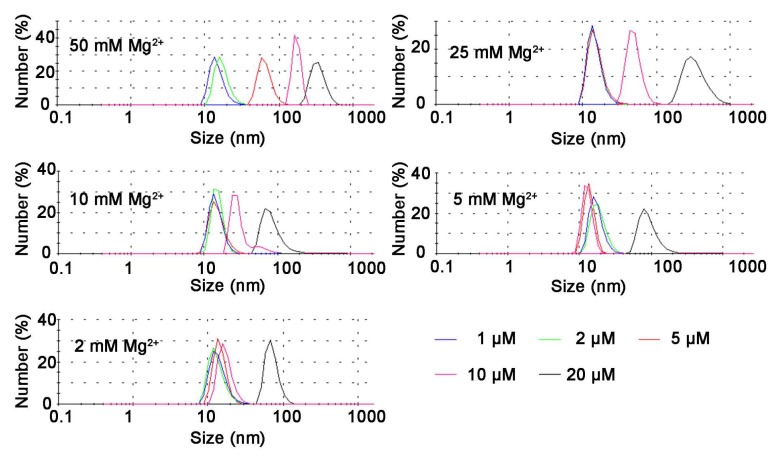
The hydrate sizes of Td (1, 2, 5, 10, 20 μM) prepared in different Mg^2+^ concentrations, analyzed by DLS.

**Figure 4 molecules-24-02091-f004:**
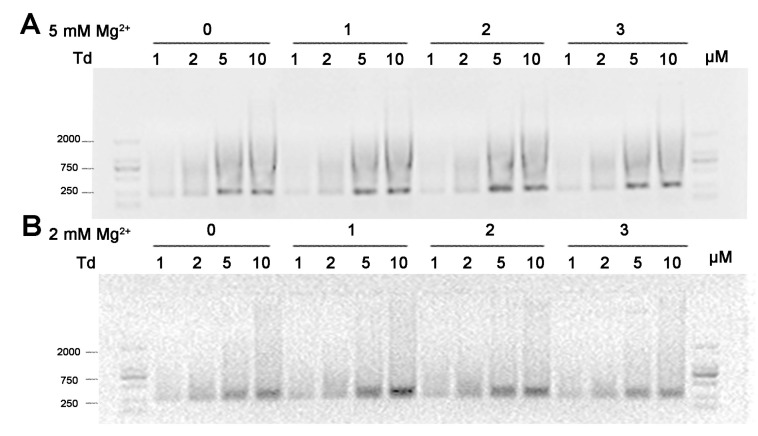
The influence of the freezing-thawing step on the stability of Td structure. Agarose gel of 1, 2, 5 or 10 μM Td prepared in buffers containing either 5 mM (**A**) or 2 mM (**B**) Mg^2+^, after repeated freeze-thawing processes. Numbers on the top line indicate the times of freeze-thawing.

**Figure 5 molecules-24-02091-f005:**
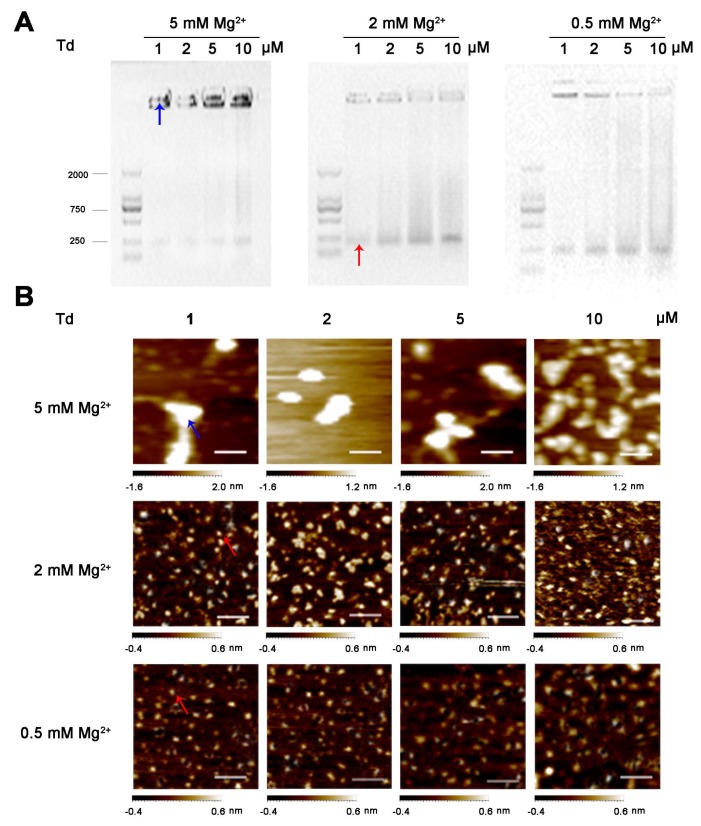
The influence of the lyophilization process on the stability of the Td structure. (**A**) Agarose gel and (**B**) AFM images of re-dissolved freeze-drying powders of 1, 2, 5 or 10 μM Td, prepared in buffers containing 5, 2 or 0.5 mM Mg^2+^. Scale bars: 100 nm. Red and blue arrows represented Td and aggregated DNA, respectively.

**Figure 6 molecules-24-02091-f006:**
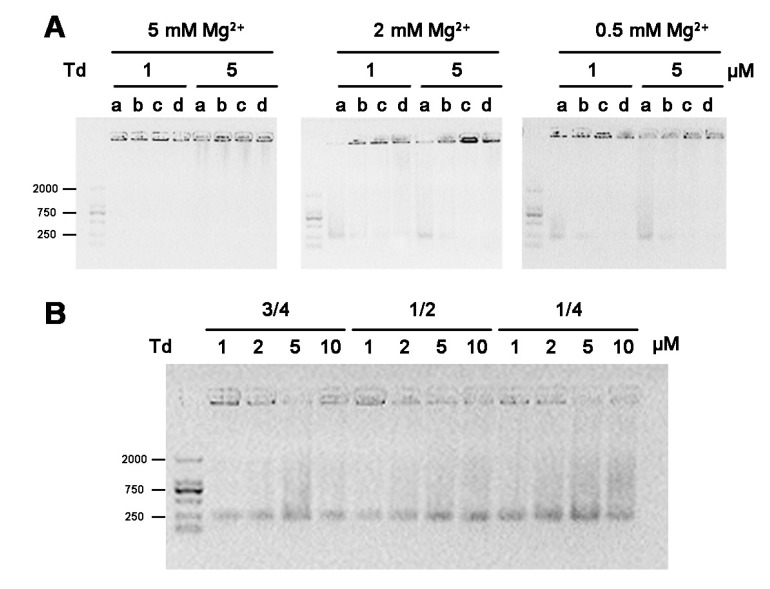
The influence of re-dissolving solvents and volumes on the stability of Td structure after freeze-drying. (**A**) Agarose gel electrophoresis of the re-dissolved 1 and 5 μM Td solutions, which were prepared in buffer containing 5, 2, or 0.5 mM Mg^2+^. “a, b, c, d” represent different re-dissolving solvents: H_2_O, PBS, MHB and DMEM, respectively. (**B**) Agarose gel electrophoresis of 1, 2, 5 or 10 μM Td re-dissolved in water, the volume of which was 3/4, 1/2 or 1/4 of the initial prepared buffer.

**Figure 7 molecules-24-02091-f007:**
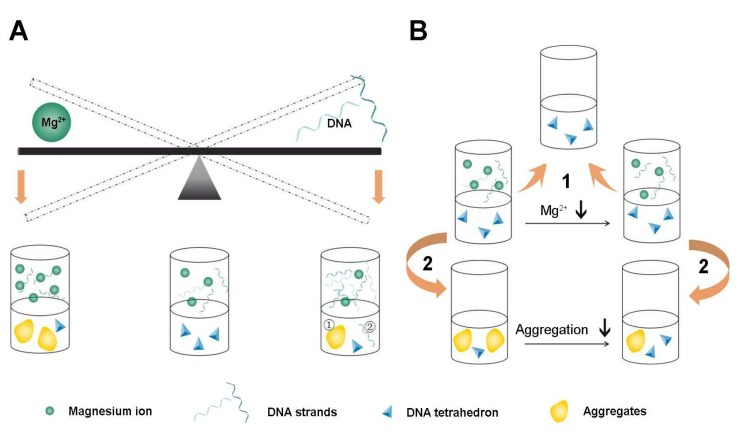
Illustrative diagram demonstrating the influence of different concentrations of Mg^2+^ and DNA on the processes of preparation, freezing-thawing and lyophilization of Td. (**A**) The balance between concentrations of Mg^2+^ and DNA strands when preparing Td. There is an appropriate range of Mg^2+^ concentrations for the successful assembly of Td. A higher concentration of Mg^2+^ (>10 mM) results in an apparent aggregation of the DNA, while a lower concentration of Mg^2+^ (<2 mM) leads to an insufficient assembly of Td. Similarly, if the concentration of DNA reaches the upper limit (20 μM), pronounced aggregation also makes the assembly fail. Therefore, successful self-assembly of the Td can only be achieved by maintaining the balance of the two ingredients. (**B**) Freeze-thawing has no influence on the structure of the formed Td (1), but lyophilization increases the risks of Td aggregation, which would be largely attenuated by reducing the concentration of Mg^2+^ in the initial prepared buffer (2).

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
