# Peer review of "Influence of Magnesium Ions on the Preparation and Storage of DNA Tetrahedrons in Micromolar Ranges"

_molecules, 2019, doi:10.3390/molecules24112091_

Round 1
Reviewer 1 Report
1. Could author rewrite the introduction part? Add more information about Mg2+-DNA interactions with sequence specific dynamic and kinetic references.
2. Figure 1B and 5B showed TD assembly at different Mg2+ concentration. The nanoparticles with a size under 100 nm but the AFM nanoparticles morphology seems far from TD. Dose the author could add one comparing AFM figures in supplementary material? They are TD assemble structure at your optimized Mg2+ condition and comparing with a control group of Na+.
Author Response
Reviewer’ s comments:
1. Could author rewrite the introduction part? Add more information about Mg2+-DNA interactions with sequence specific dynamic and kinetic references.
Answer:Thank you for your suggestions. In the revised manuscript, the introduction part has been rewritten as follows:
“Nowadays, DNA nanostructure has caught a lot of attention because of its magical properties, such as flexible sizes and shapes, excellent biocompatibility, and more possibilities to decorate precisely. Stabilized by strong hydrogen bonds, DNA nanostructure can self-assemble to designed shapes easily, causing its application in diagnosis, imaging or drug delivery widely[1]. Among these nanoparticles, DNA tetrahedron (Td), which has high yield and rigid structure, is used widely[2]. It has been reported that Td can be internalized by cell even without transfection reagent[3], based on which scientists designed various smart systems to deliver CpG motifs[4], siRNA[5] or small molecular drugs[6] and detect mRNA in living cells[7]. However, all the existing research findings were limited in laboratory. There are still many problems needing to be deeply explored for the further development of Td, including the factors influencing Td assembling process and stability during storage.
Previous published data showed that the types and concentrations of cations in the solution are the key factors affecting DNA structure stability. Compared with monovalent cations, divalent cations, such as Mg2+, have stronger interactions with DNA and greater influence on DNA structure[8]. There are two types of interactions between DNA and Mg2+: electrostatic force (phosphate groups) and chemical bonds (like hydrogen bond and coordinate bond)[9]. In solvent, Mg2+ have a coordination shell with six water molecules, which can bind to O6 of DNA to form hydrogen bonds. And in some cases, coordinate water molecules are replaced by N7 and O6 sites of guanine of DNA, forming coordination bonds between DNA and Mg2+. The effects of magnesium can be considered as a ‘protective effect’ of DNA, not only decreasing the intramolecular repulsion in DNA, but also making DNA duplex more rigid[10]. Importantly, it has been confirmed that amounts of magnesium ions affect DNA structure. DNA is stabilized in B conformation when Mg2+ is at a modest concentration, but when the concentration is extremely high, DNA structure exhibits a more compact form[11].
Therefore, the influence of Mg2+ on DNA nanostructure in applicable conditions has been paid attention. It is reported that low concentration of Mg2+ (≤1 mM) is detrimental to DNA nanostructures stability[12]. However, Keller’s group proposed that high Mg2+ concentrations are not necessary for maintaining their stability if an optimal buffer is chosen based on different forms of DNA nanostructure[13]. Furthermore, there are other ways to ensure DNA nanostructure integrity against the impacts of Mg2+, such as oligolysine-coated protection[14]. But the role of Mg2+ in DNA nanostructure preparation and storage (such as freezing-thawing and lyophilization) is not investigated.
In this study, we synthesized Td at different Mg2+ concentrations and clarified the influence of Mg2+ on Td preparation and storage, which provided some enlightenment to improve yields of other DNA nanostructures.”
2. Figure 1B and 5B showed TD assembly at different Mg2+ concentration. The nanoparticles with a size under 100 nm but the AFM nanoparticles morphology seems far from TD. Dose the author could add one comparing AFM figures in supplementary material? They are TD assemble structure at your optimized Mg2+ condition and comparing with a control group of Na+.
Answer:Thank you for your questions.
1)In Figure 1B, the products of Td assembly were showed by AFM. 1, 2, 5, 10 μM Td prepared in 5 mM Mg2+ and 1, 2 μM Td prepared in 50 mM Mg2+ were assembled successfully, and the sizes of those Td were about 10 nm. But when we were trying to prepare 20 μM Td in 5 mM Mg2+ or to prepare 5, 10, 20 μM Td in 50 mM Mg2+, DNA aggregation with larger size than 100 nm happened, as observed by AFM results. In Figure 5B, Td prepared in buffer with different Mg2+ concentration were lyophilizated, and then resolved in H2O to observe their structure by AFM. We found that Td prepared at 5 mM Mg2+ happened aggregation after lyophilization, so larger particles in AFM imaging were presented.
To display the results more clearly, we provided AFM images with higher resolutions and added some arrows in the revised manuscript. Red and blue arrows represented successfully-assembled Td and aggregated DNA , respectively.
2)Your suggestion enlightened us a lot, so we characterized the Td assembled structure in 2 mM Na+ (this chosen concentration is based on the results that maximum yield of Td could be achieved in 2 mM Mg2+). The electrophoresis results (Figure R1) showed that bands exhibited faster mobility than Td, indicating most of DNA strands were free rather than assembled into Td. Then, when imaging 1 μM products formed in 2 mM Na+ by AFM (Figure R1C), we could find nothing but the surface of mica. There are two reasons explaining this phenomenon: Firstly, most of the products prepared in 2 mM Na+ were free strands, only 64 bp, which were too difficult to be detected by AFM at our experimental conditions; Secondly, magnesium ions play an important role in strengthening the interactions between DNA and mica, but Na+ is not as good as Mg2+ to fix DNA at 2 mM concentration.
PS: Figure R1 is presented in the attached file.

Reviewer 2 Report
This paper reports the preparation and enrichment of DNA tetrahedron by investigating multiple influencing parameters such as Mg ion concentration, DNA concentration, freezing-thawing process, lyophilized treatment, and re-dissolving solvents and so on. However, this reviewer cannot recommend it publishing in Molecules.
The novelty and significance of this paper is not qualified for publishing in Molecules.
The introduction of this paper is one of the shortest introduction I have ever read. Unfortunately, it doesn’t covey the sufficient information that an introduction section should address. I would suggest the authors expanding their introduction by incorporating a more comprehensive review on the relevant field and emphasis on the novelty and importance of current work.
In the first paragraph of results, please add relevant references when referring to “based on most previous studies”.
Why the increase of Td concentration dose not result in deeper color of bands in Figure 2 A, like in 2mM Mg samples? For example, since the 2 uM and 10 uM samples have comparable yields, the latter is expected to exhibit darker bands than the former.
How DLS can provide a measurement resolution with four significant figures, like 12.36 nm? Same issue to the yield analysis in Table S3.
How to analyze the yield of Td should be detailed in the main text. Gray scanning is unspecific.
More relevant references in the field should be cited.
Author Response
Reviewer’ s comments:
1. The introduction of this paper is one of the shortest introduction I have ever read. Unfortunately, it doesn’t covey the sufficient information that an introduction section should address. I would suggest the authors expanding their introduction by incorporating a more comprehensive review on the relevant field and emphasis on the novelty and importance of current work.
Answer:Thank you for your suggestions. In the revised manuscript, the introduction part has been rewritten as follows:
“Nowadays, DNA nanostructure has caught a lot of attention because of its magical properties, such as flexible sizes and shapes, excellent biocompatibility, and more possibilities to decorate precisely. Stabilized by strong hydrogen bonds, DNA nanostructure can self-assemble to designed shapes easily, causing its application in diagnosis, imaging or drug delivery widely[1]. Among these nanoparticles, DNA tetrahedron (Td), which has high yield and rigid structure, is used widely[2]. It has been reported that Td can be internalized by cell even without transfection reagent[3], based on which scientists designed various smart systems to deliver CpG motifs[4], siRNA[5] or small molecular drugs[6] and detect mRNA in living cells[7]. However, all the existing research findings were limited in laboratory. There are still many problems needing to be deeply explored for the further development of Td, including the factors influencing Td assembling process and stability during storage.
Previous published data showed that the types and concentrations of cations in the solution are the key factors affecting DNA structure stability. Compared with monovalent cations, divalent cations, such as Mg2+, have stronger interactions with DNA and greater influence on DNA structure[8]. There are two types of interactions between DNA and Mg2+: electrostatic force (phosphate groups) and chemical bonds (like hydrogen bond and coordinate bond)[9]. In solvent, Mg2+ have a coordination shell with six water molecules, which can bind to O6 of DNA to form hydrogen bonds. And in some cases, coordinate water molecules are replaced by N7 and O6 sites of guanine of DNA, forming coordination bonds between DNA and Mg2+. The effects of magnesium can be considered as a ‘protective effect’ of DNA, not only decreasing the intramolecular repulsion in DNA, but also making DNA duplex more rigid[10]. Importantly, it has been confirmed that amounts of magnesium ions affect DNA structure. DNA is stabilized in B conformation when Mg2+ is at a modest concentration, but when the concentration is extremely high, DNA structure exhibits a more compact form[11].
Therefore, the influence of Mg2+ on DNA nanostructure in applicable conditions has been paid attention. It is reported that low concentration of Mg2+ (≤1 mM) is detrimental to DNA nanostructures stability[12]. However, Keller’s group proposed that high Mg2+ concentrations are not necessary for maintaining their stability if an optimal buffer is chosen based on different forms of DNA nanostructure[13]. Furthermore, there are other ways to ensure DNA nanostructure integrity against the impacts of Mg2+, such as oligolysine-coated protection[14]. But the role of Mg2+ in DNA nanostructure preparation and storage (such as freezing-thawing and lyophilization) is not investigated.
In this study, we synthesized Td at different Mg2+ concentrations and clarified the influence of Mg2+ on Td preparation and storage, which provided some enlightenment to improve yields of other DNA nanostructures.”
2. In the first paragraph of results, please add relevant references when referring to “based on most previous studies”.
Answer:Thank you for your suggestions. The relevant references were added and labeled in blue in the first paragraph of results. [4] doi:10.1021/nn202774x. (50 mM Mg2+) [12] doi:10.1039/c6an00241b. (50 mM Mg2+) [13] doi:10.1002/smll.201101804. (5 mM Mg2+) [14] doi:10.2147/Ijn.S132929. (5 mM Mg2+)
3. Why the increase of Td concentration dose not result in deeper color of bands in Figure 2 A, like in 2mM Mg samples? For example, since the 2 uM and 10 uM samples have comparable yields, the latter is expected to exhibit darker bands than the former.
Answer:Thank you for your questions. This phenomenon may result from over-exposure during our experimental process. As for this question, this part of experiments was repeated and the gels were imaged with appropriate exposure time and contrast ratio in order to exhibit the results more clearly. Furthermore, the arrows were added to point the specific bands. The revised Figure 2 as follows. Red, blue and black arrows represented successfully-assembled Td, aggregated DNA and free DNA strands, respectively.
4. How DLS can provide a measurement resolution with four significant figures, like 12.36 nm? Same issue to the yield analysis in Table S3.
Answer:Thank you for your questions.
1) As for DLS results, we used the Malvern Zetasizer Nano, which can provide a result with four significant figures. There are some data presented in other articles in which Malvern Zetasizer Nano was used to measure sizes.
J Control Release. 2016 Dec 10;243:121-131. (The figure is presented in the attached file.)
Biomaterials. 2013 Jul;34(21):5226-35.
“The sizes of the DNA tetrahedron and the DNA duplex were 8.89 nm and 4.19 nm, respectively.”
2)As for the Table S3, thank you for your suggestions very much. It’s our mistake to neglect the accuracy of gray scanning, and according to lots of articles which used gray scanning, the estimated yields were exhibited more appropriately with only integers.
5. How to analyze the yield of Td should be detailed in the main text. Gray scanning is unspecific.
Answer:Thank you for your suggestions. We are sorry to not detail the methods and use words exactly. Gray scanning is not a specific way to analyze the yields of Td, but a way to estimate the yields approximately. Therefore, we change the “analyze” into “estimate” in our revised manuscript to express this meaning accurately. And then, we added the method in Methods 4.2.6 as follows, “4.2.6. Estimation of Td yield. Using software ImageJ, band intensity of Td was compared with the total intensity of corresponding lane. This ratio was determined as the Td yield, which was obtained from three independent experiments.”
This method of estimating the yields of DNA nanostructure is based on previous articles.
1) Science. 2005 Dec 9;310(5754):1661-5. As mentioned, “To quantify yields, tetrahedra were run on a 6% PAGE gel (19:1 acrylamide:bisacrylamide mixture) with 1× TAE buffer and stained with SYBR Gold (Molecular Probes). Band intensities were quantified with TotalLab (Nonlinear Dynamics).Yields were estimated to be >95%, >85% for tetrahedra formed from 0.05 µM, 0.2 µM component oligonucleotides respectively.”
2) J Am Chem Soc. 2007 Jun 6;129(22):6992-3. As mentioned, “The product was run on a 6% native gel (acrylamide:bisacrylamide ratio: 19:1) and stained with SYBR Gold (Molecular Probes/Invitrogen). Using a Pharos FX Plus Molecular Imager (Bio-Rad) and Quantity One analysis software (Bio-Rad) we have determined the ratio between the intensity of the bipyramid band and the total intensity in the lane (minus the background). ”
3) Nat Commun. 2012;3:1103. As mentioned, “the yield of folding was also best for the object designed according to rule 4 with negligible by-products and approaches 100%, as measured by product band intensity versus total lane intensity excluding the excess staple band.”
6. More relevant references in the field should be cited.
Answer:Thank you for your suggestions. In revived manuscript, more relevant references were cited in the introduction part to provide more information in the field.
PS: The figures are presented in the attached file.

Reviewer 3 Report
Here Hu et al explored the importance of regulating Mg2+ ion for the assembly of DNA tetrahedrons (Td). The authors identified a range of Mg2+ ion crucial for sufficient Td yield without causing significant aggregations. The authors also evaluated the effect of Mg2+ ion towards Td freezing and lyophilization processes. I recommend the publication of this article to Molecules, subject to the following remarks:
- Td yield was calculated by evaluating band intensity at designated size? In this case, I believe the yield would be slightly misleading. In the case of DNA aggregation from Mg2+, we cannot guarantee it consisted of unassembled or assembled Td. Hope the authors can clarify this.
- The authors mention: "freeze-dried dissolving is a practicable way to concentrate Td as long as they are prepared from initial solutions with proper Mg2+ intensity". Please clarify how much concentrated it can be, and under which Mg2+ condition?
- Related to the comment above, do the author suggest assembling Td at low Mg2+, sacrificing yield but coupling it with subsequent lyophilization to concentrate, or assembling it directly at proper Mg2+ for maximum yield?
- Does Mg2+ concentration still matter following Td assembly (e.g. during its subsequent bio-applications)?
- Resolution of AFM images and some blot images can be improved.
- I suggest the authors to label precise band regions which correspond to unassembled oligos, assembled Tds and aggregated Tds.
Author Response
Reviewer’ s comments:
1. Td yield was calculated by evaluating band intensity at designated size? In this case, I believe the yield would be slightly misleading. In the case of DNA aggregation from Mg2+, we cannot guarantee it consisted of unassembled or assembled Td. Hope the authors can clarify this.
Answer:Thank you for your suggestions. Gel and AFM imaging are the most common ways to characterize DNA nanoparticles, based on which we speculate the DNA products structure. As for Td yields, we are sorry to not detail the methods and use words exactly. Gray scanning is not a specific way to analyze the yields of Td, but a way to estimate the yields approximately. Therefore, we change the “analyze” into “estimate” in our revised manuscript to express this meaning accurately. And then, we added the method in Methods 4.2.6 as follows, “4.2.6. Estimation of Td yield. Using software ImageJ, band intensity of Td was compared with the total intensity of corresponding lane. This ratio was determined as the Td yield, which was obtained from three independent experiments.”
This method of estimating the yields of DNA nanostructure is based on previous articles. 1) Science. 2005 Dec 9;310(5754):1661-5. As mentioned, “To quantify yields, tetrahedra were run on a 6% PAGE gel (19:1 acrylamide:bisacrylamide mixture) with 1× TAE buffer and stained with SYBR Gold (Molecular Probes). Band intensities were quantified with TotalLab (Nonlinear Dynamics).Yields were estimated to be >95%, >85% for tetrahedra formed from 0.05 µM, 0.2 µM component oligonucleotides respectively.”
2) J Am Chem Soc. 2007 Jun 6;129(22):6992-3. As mentioned, “The product was run on a 6% native gel (acrylamide:bisacrylamide ratio: 19:1) and stained with SYBR Gold (Molecular Probes/Invitrogen). Using a Pharos FX Plus Molecular Imager (Bio-Rad) and Quantity One analysis software (Bio-Rad) we have determined the ratio between the intensity of the bipyramid band and the total intensity in the lane (minus the background). ”
3) Nat Commun. 2012;3:1103. As mentioned, “the yield of folding was also best for the object designed according to rule 4 with negligible by-products and approaches 100%, as measured by product band intensity versus total lane intensity excluding the excess staple band.”
2. The authors mention: "freeze-dried dissolving is a practicable way to concentrate Td as long as they are prepared from initial solutions with proper Mg2+ intensity". Please clarify how much concentrated it can be, and under which Mg2+ condition?
Answer:Thank you for your suggestion. We have revised this sentence as follows, “freeze-dried dissolving is a practicable way to concentrate Td as long as they are prepared in initial solutions with Mg2+ less than 2 mM. ”
3. Related to the comment above, do the author suggest assembling Td at low Mg2+, sacrificing yield but coupling it with subsequent lyophilization to concentrate, or assembling it directly at proper Mg2+ for maximum yield?
Answer:Thank you for your question. In our manuscript, we firstly assembled Td in different concentration Mg2+, and found that the yields of Td were highest when Mg2+ was 2 or 5 mM. But when Td were lyophilizated, aggregation happened in Td prepared in 5 mM Mg2+ but not 2 mM, indicating less Mg2+ are better for Td structure during the lyophilization. Therefore, we suggested that if you want to get Td with yield as high as possible, you should assemble it in 2 mM Mg2+ and then concentrate it by lyophilization.
4. Does Mg2+ concentration still matter following Td assembly (e.g. during its subsequent bio-applications)?
Answer:Thank you for your question. It is true that Mg2+ concentration still matter following Td assembly. It has been reported that low Mg2+ concentrations (≤1 mM) damage the stability of DNA origami. And in blood and tissue culture media, Mg2+ concentrations are lower than 1 mM. Therefore, the sensitivity of DNA origami to cation depletion could influence its application. In order to solve this problem, adjusting the Mg2+ concentration in buffer is a simple way in vitro. When applied in vivo, DNA origami could be coated by capsid proteins, poly(2-dimethylaminoethylmethacrylate) (PDMAEMA)-based polymers or oligolysine to enhance its stability in physiological conditions. (ACS Nano. 2014 Sep 23;8(9):8765-75. ; Angew Chem Int Ed Engl. 2018 Jul 20;57(30):9470-9474. ; ACS Nano. 2014 Sep 23;8(9):8765-75.; Nat Commun. 2017 May 31;8:15654.; Nano Lett. 2014;14(4):2196-200.; Nanoscale. 2016 Jun 2;8(22):11674-80. ),
5. Resolution of AFM images and some blot images can be improved.
Answer:Thank you for your suggestions. We have provided images with higher resolutions in revised manuscript.
6. I suggest the authors to label precise band regions which correspond to unassembled oligos, assembled Tds and aggregated Tds.
Answer:Thank you for your suggestion. We have added some arrows in figures to exhibit results more clearly. Red, blue and black arrows represented successfully-assembled Td, aggregated DNA and free DNA strands, respectively.

Round 2
Reviewer 1 Report
Thank you for answer my two questions directly.
Author Response
Thank you very much.
Reviewer 2 Report
1. While the revised manuscript has addressed most of my concerns, I still insist on my viewpoints in the first round of reviewing that the novelty and significance of this paper is not qualified for publishing in Molecules. This comment wasn’t even mentioned in the response letter. However, the final decision should rely on the opinions of editor.
2. The authors didn’t address my following comments in the first round. “The authors should address the significance of their experiments designed for investigating the influence of freezing-thawing, and lyophilized treatment, and dissolving freeze-drying powder in different solvents. The readers will be more interested in why these experiments are designed, compared with experimental results themselves. “
3. In the revised introduction, the authors claim "However, all the existing research findings were limited in laboratory. " This argument makes one confused. It sound like the current work is addressing an issue beyond laboratory scale. Actually, it is not the case.
Author Response
1. While the revised manuscript has addressed most of my concerns, I still insist on my viewpoints in the first round of reviewing that the novelty and significance of this paper is not qualified for publishing in Molecules. This comment wasn’t even mentioned in the response letter. However, the final decision should rely on the opinions of editor.
Answer: Thank you for your question. We are so sorry to miss the comment in our previous response, so we elaborate the significance and novelty of this paper as follows:
Significance:
DNA nanostructure has attracted a lot of attention because of its special properties, such as flexible size and shape, excellent biocompatibility, and increased possibilities to decorate precisely, leading to its wide application in diagnosis, imaging or drug delivery (Chemical Society reviews 2016, 45, 4199-4225). However, its applications are hindered by one challenge: DNA nanostructure is vulnerable to environmental changes, such as Mg2+, fetal bovine serum or DNase. Mg2+ can interact with DNA by electrostatic force and chemical bonds, which is important to DNA nanostructure stability. It is reported that a low concentration of Mg2+ (≤1 mM) after self-assembly in application conditions is detrimental to the structural integrity of DNA nanostructures (ACS Nano 2014, 8, 8765-8775). But the role of Mg2+ in DNA nanostructure preparation and storage (such as freezing-thawing and lyophilization) has not yet been investigated. Therefore, this paper aims to elaborate the importance of Mg2+ in the process of preparation and storage, and obtain the optimum conditions, laying a foundation to subsequent application researches.
Novelty:
1) For the first time, we found that 2-5 mM were the optimum concentrations of Mg2+ for preparing Td in micromolar ranges. In lower concentrations, most of DNA are free single strands which can not self-assemble, while in higher concentrations, DNA occurred aggregation.
2) For the first time, we investigated the feasibility of storing Td by freezing-thawing and lyophilization and clarified that decreasing Mg2+ strength was a way to ensure the structural integrity of Td in lyophilization and water was the best solvent for solving the lyophilized powder of Td, which provides insights for other DNA nanostructures’ storage.
2. The authors didn’t address my following comments in the first round. “The authors should address the significance of their experiments designed for investigating the influence of freezing-thawing, and lyophilized treatment, and dissolving freeze-drying powder in different solvents. The readers will be more interested in why these experiments are designed, compared with experimental results themselves.”
Answer: Thank you for your question. We have explained the significance in our revised manuscript: “However, the role of Mg2+ in DNA nanostructure preparation and storage (such as freezing-thawing and lyophilization) has not yet been investigated, which is important to improve experimental reproducibility or reducing batch variations by preparing one batch Td used in multiple rounds of experiments for a longtime.” and “Because PBS, Mueller-Hinton broth (MHB) and Dulbecco's modified eagle medium (DMEM) are commonly-used biological buffers for cellular or bacterial culture, so we investigated whether lyophilizated powder of Td could be resolved in these buffers directly.”
3. In the revised introduction, the authors claim "However, all the existing research findings were limited in laboratory. " This argument makes one confused. It sound like the current work is addressing an issue beyond laboratory scale. Actually, it is not the case.
Answer: Thank you for your question. It was our expressive fault in previous revised manuscript. Therefore, we deleted this sentence and polished our manuscript.